# ADVERSARIAL SAMPLING AND PURIFICATION WITH DENOISING DIFFUSION CODEBOOK MODELS

## ABSTRACT

Diffusion models have demonstrated superior performance in both unrestricted adversarial attacks and adversarial purification. However, adversarial guidance can distort the benign sampling process, steering diffusion models toward adversarial rather than benign data distributions, thereby degrading generation performance. Meanwhile, adversarial purification struggles to isolate adversarial influence from the diffusion process, limiting its ability to recover the true benign sample. Recently, denoising diffusion codebook models have introduced a novel sampling paradigm, replacing random noise with selections from a predefined codebook. This enables an adversarially isolated sampling mechanism for diffusion models. In this paper, we propose novel frameworks for adversarial sampling and purification based on diffusion codebook sampling. Our adversarial sampling constructs adversarial examples by selectively drawing from the Gaussian noise codebook, while our purification leverages implicit guidance to suppress adversarial influence and restore benign samples. Furthermore, we introduce several enhancements to strengthen defense performance. Extensive experiments demonstrate that our method consistently outperforms state-of-the-art approaches in both adversarial sampling and purification, offering a promising direction for advancing the adversarial robustness of diffusion models.

## 1 INTRODUCTION

With their strong generative capability and denoising-like reverse process, diffusion models have shown remarkable defense potential against adversarial attacks through adversarial purification (Nie et al., 2022; Song et al., 2024). At the same time, they can also be exploited as powerful attackers against deep learning models (Chen et al., 2023c;b). Consequently, the adversarial robustness of diffusion models has attracted increasing attention from the research community.

Denoising diffusion codebook models (DDCMs) (Ohayon et al., 2025) introduce a novel generative approach by replacing Gaussian noise sampling in the diffusion process with fixed selections of noise samples from pre-defined codebooks. This mechanism not only enables diffusion models to integrate with large language models for multimodal image tasks but also raises new challenges in adversarial robustness due to their codebook sampling during the reverse generation process.

Most existing adversarial attacks using diffusion models incorporate adversarial losses into the reverse generation process to craft adversarial examples, typically formulated as $\hat{x}_t = x_t + \nabla\mathcal{L}$. Such modification shifts the data distribution of the benign data from $x_t \sim \mathcal{N}(\mu, \sigma^2 \mathbf{I})$ to an adversarial distribution $\hat{x}_t \sim \mathcal{N}(\mu + \nabla\mathcal{L}, \sigma^2 \mathbf{I})$. Therefore, existing adversarial attacks may generate out-of-distribution data of low perceptual quality that are more easily defended.

Diffusion-based adversarial purification leverages the denoising process to remove adversarial perturbations. For example, DiffPure (Nie et al., 2022) proved that for clean data sampled from clean distribution $x_t \sim p(x)$ and adversarial data from adversarial distribution $\hat{x} \sim q(t)$, we have $\frac{\partial}{\partial t} D_{KL}(p_t || q_t) \leq 0$, where the equality holds only at the beginning timestep. However, existing defenses must carefully tune $t^*$ to balance the defense performance and image consistency. Other works, such as (Wang et al., 2022), apply guided diffusion from benign generation process with guidance from the adversarial examples to purify adversarial examples, but this guidance may inadvertently shift the distribution toward the adversarial distribution, weakening robustness. Achieving strong and stable diffusion-based purification therefore remains a significant challenge.

In this paper, we conduct a systematic study of the adversarial capabilities of denoising diffusion codebook models, addressing both adversarial attack and defense perspectives. For practicality, our focus is on designing stronger defenses over the attack performance. From the attacker's side, we generate adversarial examples by conditionally sampling noise entries from the benign codebook according to target labels. Unlike gradient-based approaches, this strategy preserves the benign data distribution of trained diffusion models with benign sampling, leading to higher image quality and stronger attacks against existing defenses. From the defender's side, we present a novel diffusion-based adversarial purification that employs purification with the whole diffusion process at $\frac{\partial}{\partial t} D_{KL}(p_t||q_t) = 0$. By selecting noise samples from the codebook using posteriors derived from adversarial examples, we effectively isolate adversarial influences without explicitly injecting guidance at each timestep. To further enhance the purification performance, we provide an effective preprocessing approach and incorporate random noise sampling to better recover the benign data distribution given adversarial examples. Comprehensive experiments across multiple datasets validate both the adversarial strength and defensive robustness of DDCMs. Our results establish a new direction for advancing adversarial robustness in diffusion models, achieving state-of-the-art performance in both attack and defense scenarios.

Our contributions are summarized as follows:

- We conduct pioneering research to investigate adversarial attacks and defenses with DD-CMs, leveraging fixed codebook-based noise sampling in pre-trained diffusion models. Our study reveals their effectiveness for both adversarial attacks and adversarial purification.

- We propose an adversarial attack method with DDCMs to craft unrestricted adversarial examples by conditional noise selection from the codebook. Unlike previous approaches, our attacks preserve the benign data distribution, yielding higher image quality and stronger resistance against defenses.

- We introduce an effective diffusion-based purification framework with DDCMs. The strength of our defense lies in adversarial-isolated sampling with benign codebook noise selection, while leveraging the full diffusion process. We also present several enhancement measurements to improve defense performance.

- We conduct extensive experiments across multiple datasets, demonstrating that codebook-based sampling provides both strong attack capability and robust defense performance, achieving state-of-the-art results in adversarial robustness with diffusion models.

## 2 PRELIMINARY AND RELATED WORKS

### 2.1 DENOISING DIFFUSION CODEBOOK MODELS

The Denoising Diffusion Probabilistic Model (DDPM) (Ho et al., 2020) learns the data distribution with a prior Gaussian distribution through a forward diffusion process and a reverse generation process. In the reverse generation process, we generate data from the data distribution $p$ with $T$ time steps. We obtain data $\boldsymbol{x}_0$ with a chain of data $\{\boldsymbol{x}_{T-1}, \ldots, \boldsymbol{x}_1\}$. At each timestep $t$, we sample the data $\boldsymbol{x}_{t-1}$ with:

$$p_\theta(\boldsymbol{x}_{t-1}|\boldsymbol{x}_t) = \mathcal{N}(\boldsymbol{x}_{t-1} : \boldsymbol{\mu}_\theta(\boldsymbol{x}_t, t), \sigma_\theta^2(\boldsymbol{x}_t, t)) \tag{1}$$

where $\boldsymbol{\mu}_\theta$ is learned by a deep learning model $\boldsymbol{\epsilon}_\theta$ parameterized by $\theta$, and $\sigma_\theta$ is a predefined parameter related to $t$ by Ho et al. (2020).

The forward diffusing process gradually adds Gaussian noise to the data sampled from a conditional distribution, which is defined as:

$$q(\boldsymbol{x}_t|\boldsymbol{x}_{t-1}) = \mathcal{N}(\boldsymbol{x}_t : \sqrt{\alpha_t}\boldsymbol{x}_{t-1}, (1 - \alpha_t)\mathbf{I}) \tag{2}$$

where $\alpha \in (0, 1)$

Equation 1 indicates that at each DDPM sampling process, we sample a Gaussian noise with a reparameterization trick from the standard Gaussian distribution. Denoising Diffusion Codebook Models (DDCMs) (Ohayon et al., 2025) replace this random noise sampling with a fixed, discrete noise codebook. Specifically, the codebook $\mathcal{C}$ stores $K$ constant noise vectors for each timestep $t$, formulated as:

$$\mathcal{C}_t = [\boldsymbol{z}_t^{(1)}, \boldsymbol{z}_t^{(2)}, ..., \boldsymbol{z}_t^{(K)}] \tag{3}$$

where each $z$ is independently sampled from $\mathcal{N}(0, \mathbf{I})$ only at the model initialization phase. Then, the reverse generation process of DDCM is defined as:

$$\boldsymbol{x}_{t-1} = \boldsymbol{\mu}_\theta(\boldsymbol{x}_t, t) + \sigma_\theta \mathcal{C}_t(k_t) \tag{4}$$

where the generation process of DDCM adopt the fixed sampled noise for the generation of $\boldsymbol{x}_0$. We can use the sequence of the index of the codebook $\{k_T, ..., k_1\}$ to represent each of the generated images from DDCM.

Given a data sample $\boldsymbol{x}$, DDCM can either reconstruct $\boldsymbol{x}$ or obtain its compressed representation by guiding the reverse generation process. At each timestep, the model selects the corresponding codebook entry aligned with $\boldsymbol{x}$, thereby encoding the sample into its discrete index sequence:

$$k_t = \underset{1,...,K}{\arg\max}\, \mathcal{C}_t(k_t) \cdot (\boldsymbol{x} - \bar{\boldsymbol{x}}_{0,t}) \tag{5}$$

where $\bar{\boldsymbol{x}}_{0,t}$ is the predicted $\boldsymbol{x}_0$ at timestep $t$:

$$\bar{\boldsymbol{x}}_{0,t} = \frac{1}{\sqrt{\bar{\alpha}_t}}(\boldsymbol{x}_t - \sqrt{1 - \bar{\alpha}_t}\boldsymbol{\epsilon}_\theta(\boldsymbol{x}_t, t)) \tag{6}$$

## 2.2 DIFFUSION-BASED ADVERSARIAL ATTACKS

To employ diffusion models for adversarial attacks, the reverse generation process is modified by incorporating an adversarial objective, as proposed in prior works (Chen et al., 2023b;a). Formally, diffusion-based adversarial attacks at timestep $t$ can be expressed as:

$$\hat{\boldsymbol{x}}_{t-1} = \boldsymbol{\mu}_\theta(\hat{\boldsymbol{x}}_t, t) + \sigma_\theta \boldsymbol{z}_t + \nabla\mathcal{L}(\bar{\boldsymbol{x}}_{0,t}, y_a) \tag{7}$$

where $\boldsymbol{z}_t \sim \mathcal{N}(0, \mathbf{I})$, $\nabla\mathcal{L}$ is the adversarial loss function defined by the attacker.

Existing attacks design various loss functions to improve the effectiveness of the generated adversarial examples $\hat{\boldsymbol{x}}$. However, these methods alter the sampling process, shifting it from $\boldsymbol{x}_t \sim \mathcal{N}(\boldsymbol{\mu}, \sigma^2\mathbf{I})$ to the adversarial distribution $\hat{\boldsymbol{x}} \sim \mathcal{N}(\boldsymbol{\mu} + \nabla\mathcal{L}, \sigma^2\mathbf{I})$.

## 2.3 DIFFUSION-BASED ADVERSARIAL PURIFICATION

Diffusion-based adversarial purification leverages the strong generative ability of diffusion models to remove perturbations from adversarial examples. Existing defenses can be broadly categorized into two classes.

Following Nie et al. (2022), one line of work applies a partial forward diffusion process to project adversarial examples back toward the benign data distribution, and then reconstructs clean samples via the reverse process. These methods aim to identify an optimal timestep $t^*$ such that:

$$\boldsymbol{x}_{t^*} = \sqrt{\alpha_{t^*}}\hat{\boldsymbol{x}} + \sqrt{1 - \alpha_{t^*}}\boldsymbol{v}_{t^*} \tag{8}$$

where $\hat{\boldsymbol{x}}$ is the adversarial example, and $\boldsymbol{v}_{t^*}$ is the random noise. Following the forward diffusion process with $t^*$ timestep, they recover the clean data $\boldsymbol{x}_0$ with the reverse generation process. We call these defenses as FAB (Forward and Backward).

Another line of adversarial purification leverages guided diffusion models. These approaches utilize the entire reverse generation process, starting from $\boldsymbol{x}_T$ sampled from a random Gaussian distribution. The difference between the adversarial example and the purified sample is incorporated as guidance throughout the reverse process. At timestep $t$, the purification can be formulated as:

$$\boldsymbol{x}_{t-1} = \boldsymbol{\mu}_\theta(\boldsymbol{x}_t, t) + \sigma_\theta \boldsymbol{z}_t \nabla_{\boldsymbol{x}} + \log p(\hat{\boldsymbol{x}}|\boldsymbol{x}_t; t) \tag{9}$$

where the guidance is defined as:

$$\nabla_{\boldsymbol{x}} \log p(\hat{\boldsymbol{x}}|\boldsymbol{x}_t; t) = -R_t \nabla_{\boldsymbol{x}_t} d(\bar{\boldsymbol{x}}_{0,t}, \boldsymbol{x}_{\text{adv}}) \tag{10}$$

where $R_t$ is the scale factor, $d(\cdot)$ is the distance function, and $\bar{\boldsymbol{x}}_{0,t}$ is the predicted $\boldsymbol{x}_0$ defined at Equation 6. We call these defenses as GB (Guided Backward).

Existing diffusion-based adversarial purification methods employ various diffusion models, such as discrete diffusion models (DDPM) or continuous diffusion models (SDE). For simplicity, we adopt DDPM sampling for the remainder of this paper.

---

**Algorithm 1** DDCM Adversarial Sampling

---

**Require:** target label for adversarial attack $y_a$, ground truth label $y$, the optimal time $t^*$, the benign data $\boldsymbol{x}_B$, target model $f$.

1: $\boldsymbol{x}_T \sim \mathcal{N}(0, \mathbf{I})$, initialize $\mathcal{C}$ with Equation 3
2: **for** $t = T, \ldots, t^*$ **do**
3:     Obtain inference of predicted $\bar{\boldsymbol{x}}_{0,t}$ from $\boldsymbol{\epsilon}_\theta(\boldsymbol{x}_t, t, y)$
4:     Codebook selection $k_t = \arg\max_{1,\ldots,K} \mathcal{C}_t(k_t) \cdot (\boldsymbol{x}_B - \bar{\boldsymbol{x}}_{0,t})$ with Equation 6
5:     Sampling data $\boldsymbol{x}_{t-1} = \boldsymbol{\mu}_\theta(\boldsymbol{x}_t, t, y) + \sigma_\theta \mathcal{C}_t(k_t)$
6: **end for**
7: **for** $t = t^*, \ldots, 1$ **do**
8:     Obtain inference of predicted $\bar{\boldsymbol{x}}_{0,t}$ from $\boldsymbol{\epsilon}_\theta(\boldsymbol{x}_t, t, y)$
9:     Adversarial codebook selection $\hat{k}_t = \arg\max_{1,\ldots,K} \mathcal{C}_t(k_t) \cdot \nabla_{\boldsymbol{x}_t} \log p_f(y_a|\bar{\boldsymbol{x}}_{0,t})$
10:     Adversarial sampling $\hat{\boldsymbol{x}}_{t-1} = \boldsymbol{\mu}_\theta(\boldsymbol{x}_t, t, y) + \sigma_\theta \mathcal{C}_t(\hat{k}_t)$
11: **end for**
12: $\boldsymbol{x}_{\text{adv}} \leftarrow \hat{\boldsymbol{x}}_0$ if $f(\hat{\boldsymbol{x}}_0) = y_a$
13: **return** $\boldsymbol{x}_{\text{adv}}$

---

## 3 METHODOLOGY

In this section, we present our adversarial sampling and purification framework using DDCMs. Our goal is to generate samples from the benign data distribution rather than the adversarial distribution, thereby fully leveraging the generative capabilities of diffusion models.

### 3.1 DIFFUSION-BASED CODEBOOK ADVERSARIAL ATTACK

Our proposed adversarial attack leverages conditional sampling in guided diffusion models to generate adversarial examples. The adversarial guidance follows the classifier-guided guidance, but replaces the classifier with the target model. Incorporating the codebook $\mathcal{C}$, the adversarial sampling process of our attack is defined as:

$$\hat{\boldsymbol{x}}_{t-1} = \boldsymbol{\mu}_\theta(\boldsymbol{x}_t, t) + \sigma_\theta \mathcal{C}_t(\hat{k}_t) \tag{11}$$

where we only modify the selection of the codebook during the reverse generation process. We select $\hat{k}_t$ according to:

$$\hat{k}_t = \arg\max_{1,\ldots,K} \mathcal{C}_t(k_t) \cdot \nabla_{\boldsymbol{x}_t} \log p_f(y_a|\bar{\boldsymbol{x}}_{0,t}) \tag{12}$$

where $f$ is the target model and $y_a$ is the target label for the adversarial attack. The theoretical proof of Equation 12 is given in the Appendix.

Nie et al. (2022) demonstrated that there exists an optimal timestep $t^*$ at which the benign and adversarial data distributions become indistinguishable. Accordingly, we introduce adversarial sampling only after $t^*$ to ensure that the initial sampling is drawn from the benign data distribution. The complete adversarial attack procedure is summarized in Algorithm 1.

Algorithm 1 employs a conditional pre-trained DDPM $\boldsymbol{\epsilon}_\theta$ and a reference benign data $\boldsymbol{x}_B$ to facilitate sampling from the benign data distribution with respect to the ground-truth label $y$. In practice, however, these components are not essential for the success of the proposed adversarial attack and can be replaced by an unconditional DDPM and random selection of $k_t$ at Line 4.

Unlike existing methods, our adversarial attack does not introduce perturbations into the benign sampling process of diffusion models or the benign data distribution. Instead, it modifies only the codebook selection principle through conditional sampling with guidance. As a result, the proposed attack achieves higher sampling quality while remaining more concealed from current diffusion-based purification methods due to its implicit adversarial sampling mechanism.

### 3.2 DIFFUSION-BASED CODEBOOK PURIFICATION

Given the benign data distribution $\boldsymbol{x}_t \sim \mathcal{N}(\boldsymbol{\mu}, \sigma^2 \mathbf{I})$ and the adversarial distribution $\hat{\boldsymbol{x}} \sim \mathcal{N}(\boldsymbol{\mu} + \nabla\mathcal{L}, \sigma^2 \mathbf{I})$, existing diffusion-based purification methods assume that adversarial examples can be

---

**Algorithm 2** EDDCM Adversarial Purification

---

**Require:** the adversarial data $\hat{x}$, the optimal time $t^*$, the blurring filter $BF$.
1: $x_T \sim \mathcal{N}(0, \mathbf{I})$, initialize $\mathcal{C}$ with Equation 3
2: $x_{\text{ref}} = BF(\hat{x})$
3: **for** $t = T, \ldots, t^*$ **do**
4:     Obtain inference of predicted $\bar{x}_{0,t}$ from $\epsilon_\theta(x_t, t)$
5:     Codebook selection $k_t = \arg\max_{1,\ldots,K} \mathcal{C}_t(k_t) \cdot (x_{\text{ref}} - \bar{x}_{0,t})$
6:     Sampling data $x_{t-1} = \mu_\theta(x_t, t) + \sigma_\theta \mathcal{C}_t(k_t)$
7: **end for**
8: **for** $t = t^*, \ldots, 1$ **do**
9:     Obtain inference of predicted $\bar{x}_{0,t}$ from $\epsilon_\theta(x_t, t)$
10:     Random codebook selection $k_t = \text{Uniform}(\{1, ..., K\})$
11:     Sampling data $x_{t-1} = \mu_\theta(x_t, t) + \sigma_\theta \mathcal{C}_t(k_t)$
12: **end for**
13: **return** $x_0$

---

recovered to their corresponding benign data. However, FAB approaches only identify an optimal timestep $t^*$, where differences between the benign and adversarial distributions still remain. In contrast, GB approaches initialize from $x_T$ where $\frac{\partial}{\partial t} D_{KL}(p_t|q_t) = 0$, but rely on guidance derived from adversarial examples. This reliance risks shifting the purification process toward the adversarial distribution itself. To address this issue, we propose a more robust purification framework that follows the GB paradigm but eliminates explicit adversarial guidance.

### 3.2.1 BASELINE PURIFICATION

Our baseline adversarial purification directly applies the reverse generation process of codebook-based diffusion sampling, defined as:

$$x_{t-1} = \mu_\theta(x_t, t) + \sigma_\theta \mathcal{C}_t(k_t) \tag{13}$$

where $k_t$ is selected as:

$$k_t = \arg\max_{1,\ldots,K} \mathcal{C}_t(k_t) \cdot (\hat{x} - \bar{x}_{0,t}) \tag{14}$$

where $\hat{x}$ is the adversarial example expected to be purified.

Our baseline idea is to recover the closest benign sample $x$ to an adversarial example $\hat{x}$ by sampling from $\mathcal{N}(\mu, \sigma^2 \mathbf{I})$. With an infinitely large codebook $\mathcal{C}$, any adversarial sample $\hat{x} \sim \mathcal{N}(\mu + \nabla\mathcal{L}, \sigma^2 \mathbf{I})$ could, in principle, be mapped back to its corresponding benign $x$ from $x_T$, where $\frac{\partial}{\partial t} D_{KL}(p_t|q_t) = 0$. In practice, however, $\mathcal{C}$ has a finite length, meaning that certain adversarial samples—particularly those with larger $\nabla\mathcal{L}$—cannot be perfectly reversed. Despite this limitation, our defense consistently outperforms existing baseline purification methods.

### 3.2.2 ENHANCED PURIFICATION

We propose a two-fold enhancement to further strengthen the defense performance of our baseline method. The objectives are (1) to isolate the influence of adversarial perturbations and (2) to improve sampling from the benign distribution without the influence from $\hat{x}$.

**Blurring Enhancement.** The perturbation budget $\delta$ plays a critical role in diffusion-based adversarial purification, as it determines the optimal timestep $t^*$ for FAB approaches and influences the effectiveness of guidance in GB approaches. Consequently, reducing the magnitude of adversarial perturbations can further enhance purification performance. To this end, we apply a predefined filter to blur the adversarial example $\hat{x}$, thereby reduce the perturbations. We formalize this claim as follows:

**Theorem 1** *Suggest the adversarial example $\hat{x}_t$ and blurred example $\hat{x}_t^{blur}$ is the input to the diffusion model at timestep $t$, the probability of obtaining the clean purified examples $x_0$ is denoted as $P(D)$ and $P(G)$, respectively. If the timestep is infinite, we have:*

$$P(G) > P(D) \tag{15}$$

Table 1: **The ASR and image quality on the ImagetNet dataset.**

| Method | ASR (%) | FID (↓) | LPIPS (↓) | BRISQUE (↓) | TRES (↑) |
|---|---|---|---|---|---|
| ResNet50 | | | | | |
| AutoAttack | 95.1 | 26.5 | 0.72 | 34.4 | 69.8 |
| DiffAttack | 92.8 | 20.5 | 0.15 | 22.6 | 67.8 |
| AdvDiff | **99.8** | 16.2 | **0.03** | 18.1 | 82.1 |
| $DDCM_{Adv}$ | 98.3 | **12.8** | 0.28 | **16.2** | **88.7** |
| Method | ASR (%) | FID (↓) | LPIPS (↓) | BRISQUE (↓) | TRES (↑) |
| ViT-B | | | | | |
| AutoAttack | 93.8 | 25.4 | 0.75 | 41.4 | 66.4 |
| DiffAttack | 90.6 | 20.7 | 0.24 | 23.5 | 67.1 |
| AdvDiff | **98.9** | 17.4 | **0.05** | 19.5 | 80.5 |
| $DDCM_{Adv}$ | 84.3 | **13.4** | 0.33 | **19.1** | **86.8** |

Theorem 1 indicates that blurred examples achieve better defense performance than non-purified adversarial input. The detailed proof is given in the Appendix.

**Sampling Enhancement.** FAB purification methods suggest that the timestep $t^*$ determines whether diffusion models can still distinguish between benign and adversarial examples during purification. Therefore, adversarial examples should not be used as guidance beyond $t^*$. To address this, we introduce a sampling enhancement that employs random sampling to further improve defense performance.

Incorporating the enhancements described above, our enhanced purification framework with DDCM is summarized in Algorithm 2.

## 4 EXPERIMENTS

### 4.1 EXPERIMENTAL SETUP

**Dataset and Target Models.** We evaluate our attacks and defenses on CIFAR-10 (Krizhevsky et al., 2009) and ImageNet (Deng et al., 2009) dataset. We select commonly used backbones for target models: WideResNet-28-10 (default target model) and WideResNet-70-16 (Zagoruyko & Komodakis, 2016) for CIFAR-10 dataset and ResNet50 (He et al., 2016) (default target model) and ViT (Dosovitskiy et al., 2020) for ImageNet dataset.

**Attack Comparisons.** We compared our adversarial sampling mainly with existing diffusion-based white-box unrestricted adversarial attacks, i.e., DiffAttack (Chen et al., 2023a) and AdvDiff (Dai et al., 2023) for comparisons. We further include several perturbation-based adversarial attacks, i.e., AutoAttack (Croce & Hein, 2020) and PGD (Madry et al., 2018). For evaluation metrics, we evaluate Attack Success Rate (ASR) and image quality under various metrics (i.e., LPIPS (Zhang et al., 2018), BRISQUE (Mittal et al., 2011), TRES (Golestaneh et al., 2022)).

**Defense Comparisons.** We evaluate our purification methods with state-of-the-art diffusion-based purification for major evaluation. For FAB approaches, we compare with Nie et al. (2022)'s Diff-Pure, Lee et al.'s (Lee & Kim, 2023), and Bai et al.'s (Bai et al., 2024). For GB approaches, we evaluate with Wang et al.'s (Wang et al., 2022) and Song et al.'s MimicDiffusion (Song et al., 2024). We implement the code with the standardized benchmark platform RobustBench (Croce et al., 2021). We also include previous adversarial training and non-diffusion purification, followed by DiffPure (Nie et al., 2022), for comprehensive discussions. For the diffusion model backbones, we select pre-trained Score SDE (Song et al., 2021b) and DDIM (Song et al., 2021a) for the CIFAR-10 dataset, and LDM (Rombach et al., 2022) for the ImageNet dataset. We use *standard accuracy* and *robust accuracy* as the evaluation metrics with the top-1 classification result.

**Implementation Details.** We utilize the pre-trained Score SDE (Song et al., 2021b) for the CIFAR-10 dataset, and LDM (Rombach et al., 2022) for the ImageNet dataset with DDPM sampler for our attack and defense methods. We follow the selection of $t^*$ as in DiffPure (Nie et al., 2022). For

Table 2: **The ASR of ResNet50 examples against defenses on the ImagetNet dataset.**

| Method | DiffPure | Lee & Kim (2023) | Bai et al. (2024) | MimicDiffusion | AdvProp |
|---|---|---|---|---|---|
| AutoAttack | 22.2 | 21.8 | 20.4 | 9.3 | 69.6 |
| DiffAttack | 30.6 | 26.3 | 27.3 | 17.4 | 85.1 |
| AdvDiff | 41.6 | 45.4 | 14.9 | 17.8 | 89.7 |
| $DDCM_{Adv}$ | **52.7** | **48.9** | **42.7** | **36.0** | **91.1** |

| Method | Adv-Inception | HGD | R&P | RS | NRP |
|---|---|---|---|---|---|
| AutoAttack | 14.6 | 20.5 | 20.6 | 38.9 | 39.4 |
| DiffAttack | 30.9 | 20.5 | 23.7 | 40.8 | 38.5 |
| AdvDiff | 19.4 | 17.8 | 17.4 | 47.6 | 45.2 |
| $DDCM_{Adv}$ | **55.9** | **49.0** | **81.2** | **83.6** | **82.4** |

blurring enhancement, we select the mean filter $[[1, 1, 1, 1, 1], [1, 1, 0, 1, 1], [1, 1, 1, 1, 1]]$ for both CIFAR-10 dataset (upscale to $256 \times 256$ resolution) and ImageNet dataset.

## 4.2 ATTACK PERFORMANCE

### 4.2.1 ATTACK ASR AND IMAGE QUALITY

The results in Table 1 demonstrate that our adversarial sampling achieves substantially higher image quality than prior attack methods. This improvement arises from sampling adversarial examples from $\mathcal{N}(\boldsymbol{\mu}, \sigma^2 \mathbf{I})$ rather than $\mathcal{N}(\boldsymbol{\mu} + \nabla \mathcal{L}, \sigma^2 \mathbf{I})$. However, employing a fixed codebook length $K$ can reduce ASR. To address this limitation, ASR can be improved by generating multiple samples, which alleviates the constraint of a fixed $K$ but comes at the expense of time efficiency. An alternative enhancement is to increase the codebook length $K$, as further discussed in the Appendix. Noted that the clean examples for AutoAttack and PGD are images generated by LDM.

### 4.2.2 ATTACK AGAINST DEFENSE

We evaluate the attack performance against a range of adversarial defenses in Table 2. For diffusion-based adversarial purification, we select four representative methods. For adversarial training–based defenses, we include AdvProp (Xie et al., 2020), Adv-Inception (Madry et al., 2018), HGD (Liao et al., 2018), R&P (Xie et al., 2018), RS (Cohen et al., 2019), and NRP (Naseer et al., 2020). The results show that, although our attack does not achieve the highest overall ASR, it attains significantly higher ASR against state-of-the-art defenses. This finding suggests that our adversarial examples are inherently closer to the benign data distribution, making them more effective in bypassing robust defense mechanisms.

## 4.3 DEFENSE PERFORMANCE

### 4.3.1 DEFENSE ACCURACY

The experimental results on the CIFAR-10 and ImageNet datasets, reported in Table 3 and Table 4, validate the effectiveness of our proposed purification method. Our baseline defense outperforms the baseline purifications of both FAB and GB approaches ( Nie et al. (2022) and Wang et al. (2022)), particularly on the high-resolution ImageNet dataset, demonstrating that diffusion-based codebook sampling can effectively isolate the influence of adversarial perturbations and achieve stronger defense performance. Moreover, our enhanced purification further improves robustness, achieving state-of-the-art robust accuracy across different datasets. A slight reduction in standard accuracy is observed with our EDDCM due to the inherent randomness in sampling. For adaptive attacks, as shown in Table 5 and Table 6, our purification leverages in-place operations with codebook selection and blurring enhancement. Since these operations have zero gradients, they prevent gradient-based optimization attacks from effectively adapting to the defense, resulting in significantly improved robustness against adaptive attacks.

Table 3: **The standard and robust accuracy against AutoAttack ($\ell_\infty = 8/255$) on the CIFAR-10 dataset.**

| Method | Standard Acc | Robust Acc |
|---|---|---|
| WideResNet-28-10 | | |
| Wu et al. (2020) | 85.36 | 59.18 |
| Gowal et al. (2021) | 87.33 | 61.72 |
| Rebuffi et al. (2021) | 87.50 | 65.24 |
| Nie et al. (2022) | 89.23 | 71.03 |
| Lee & Kim (2023) | 90.16 | 70.47 |
| Bai et al. (2024) | 91.41 | 82.81 |
| Wang et al. (2022) | 84.85 | 71.18 |
| Song et al. (2024) | 92.10 | 75.45 |
| DDCM$_{Pure}$ | **93.85** | 71.38 |
| EDDCM$_{Pure}$ | 90.31 | **86.40** |
| WideResNet-70-16 | | |
| Rebuffi et al. (2021) | 88.54 | 64.46 |
| Gowal et al. (2021) | 88.74 | 66.60 |
| Nie et al. (2022) | 91.04 | 71.84 |
| Lee & Kim (2023) | 90.43 | 66.06 |
| Bai et al. (2024) | 92.97 | 82.81 |
| Song et al. (2024) | 93.25 | 76.60 |
| DDCM$_{Pure}$ | **94.74** | 75.80 |
| EDDCM$_{Pure}$ | 92.65 | **87.55** |

Table 4: **The standard and robust accuracy against AutoAttack ($\ell_\infty = 8/255$) on the ImageNet dataset.**

| Method | Standard Acc | AutoAttack Acc |
|---|---|---|
| ResNet50 | | |
| Croce et al. (2021) | 62.56 | 31.06 |
| Wong et al. (2020) | 55.62 | 26.95 |
| Salman et al. (2020) | 64.02 | 37.89 |
| Bai et al. (2021) | 67.38 | 35.51 |
| Nie et al. (2022) | 68.22 | 43.89 |
| Lee & Kim (2023) | **70.74** | 46.70 |
| Bai et al. (2024) | 70.41 | 45.59 |
| Wang et al. (2022) | 70.17 | 60.28 |
| Song et al. (2024) | 66.92 | 61.53 |
| DDCM$_{Pure}$ | 69.47 | 61.52 |
| EDDCM$_{Pure}$ | 68.82 | **65.03** |
| ViT-B | | |
| Nie et al. (2022) | 71.25 | 50.14 |
| Bai et al. (2024) | 73.80 | 52.87 |
| Song et al. (2024) | 70.62 | 62.58 |
| DDCM$_{Pure}$ | **74.21** | 62.43 |
| EDDCM$_{Pure}$ | 73.05 | **66.17** |

Table 5: **The robust accuracy against adaptive attacks, i.e., BPDA ($\ell_\infty = 8/255$) and PGD+EOT ($\ell_\infty = 8/255$) on the CIFAR-10 dataset with WideResNet-28-10 as the target model.**

| Method | BPDA Acc | PGD+EOT Acc |
|---|---|---|
| Nie et al. (2022) | 81.56 | 46.84 |
| Lee & Kim (2023) | 83.44 | 55.82 |
| Bai et al. (2024) | 81.94 | 72.68 |
| Wang et al. (2022) | 78.40 | 39.51 |
| Song et al. (2024) | 76.45 | 68.20 |
| DDCM$_{Pure}$ | 77.51 | 71.87 |
| EDDCM$_{Pure}$ | **86.28** | **80.43** |

Table 6: **The robust accuracy against PGD ($\ell_\infty = 4/255$) and PGD+EOT ($\ell_\infty = 4/255$) on the ImageNet dataset.**

| Method | PGD Acc | PGD+EOT Acc |
|---|---|---|
| Wong et al. (2020) | 26.24 | 30.51 |
| Salman et al. (2020) | 34.96 | 38.62 |
| Bai et al. (2021) | 40.27 | 43.42 |
| Nie et al. (2022) | 42.88 | 38.71 |
| Lee & Kim (2023) | 46.31 | 42.15 |
| Bai et al. (2024) | 47.48 | 45.87 |
| Song et al. (2024) | 62.16 | 52.66 |
| DDCM$_{Pure}$ | 62.76 | 59.21 |
| EDDCM$_{Pure}$ | **67.32** | **64.45** |

### 4.3.2 TIME EFFICIENCY

We evaluate the time efficiency of different diffusion-based purification methods for processing a single image, as reported in Table 7 and Table 8. The results indicate that GB approaches suffer from significantly reduced efficiency due to the need for gradient computation. In contrast, our method achieves substantially better efficiency by eliminating the gradient calculation requirement, suggesting a promising alternative guidance for GB approaches. On smaller datasets such as CIFAR-10, the overhead of codebook selection can be relatively higher because of the small image resolution. However, on larger datasets, the codebook selection process is considerably faster than gradient-based operations, highlighting the scalability advantage of our method.

### 4.3.3 COMBINING WITH EXISTING DEFENSES

Our proposed DDCM purification can be seamlessly integrated with existing purification methods based on DDPM or SDE samplers. The experimental results, presented in Table 9, show that replacing the noise selection in the reverse generation step consistently enhances the performance of existing defenses. This demonstrates the flexibility and practicality of DDCM as a plug-and-play

Table 7: **The time cost against AutoAttack ($\ell_\infty = 8/255$) on the CIFAR-10 dataset.**

| Method | Robust Acc | Time Cost(s) |
|---|---|---|
| Nie et al. (2022) | 71.03 | **0.6** |
| Lee & Kim (2023) | 70.47 | 1.8 |
| Bai et al. (2024) | 70.47 | 2.1 |
| Wang et al. (2022) | 71.18 | 2.2 |
| Song et al. (2024) | 75.45 | 2.6 |
| DDCM$_{Pure}$ | 71.38 | 3.6 |
| EDDCM$_{Pure}$ | **86.40** | 2.8 |

Table 8: **The time cost against AutoAttack ($\ell_\infty = 8/255$) on the ImageNet dataset.**

| Method | Robust Acc | Time Cost(s) |
|---|---|---|
| Nie et al. (2022) | 43.89 | **13.3** |
| Lee & Kim (2023) | 46.70 | 32.4 |
| Bai et al. (2024) | 45.59 | 51.8 |
| Wang et al. (2022) | 60.28 | 62.8 |
| Song et al. (2024) | 61.53 | 146.1 |
| DDCM$_{Pure}$ | 61.52 | 24.7 |
| EDDCM$_{Pure}$ | **65.03** | 22.8 |

Table 9: **The integration of proposed purification with existing methods on the CIFAR-10 dataset.**

| Method | Robust Acc |
|---|---|
| Nie et al. (2022) | 71.03 |
| +DDCM$_{Pure}$ | 73.24 |
| +EDDCM$_{Pure}$ | **81.74** |
| Wang et al. (2022) | 71.18 |
| +DDCM$_{Pure}$ | 76.05 |
| +EDDCM$_{Pure}$ | **83.25** |

Figure 1: **The ablation study of parameters with EDDCM$_{Pure}$ and ResNet50 against AutoAttack ($\ell_\infty = 8/255$) on the ImageNet dataset.**

improvement that only requires modifications to the noise sampling of the underlying sampling algorithms. Moreover, this combination further strengthens robustness against adaptive attacks, as the in-place operations employed in our method introduce gradient obfuscation that hinders adversarial optimization.

### 4.4 ABLATION STUDY

**Selection of $K$.** The length of $\mathcal{C}$ has a significant impact on the performance of our purification method. The parameter $K$ controls the search space for noise selection, directly influencing both the quality of the recovered images and the effectiveness of adversarial purification. As illustrated in Figure 1, increasing $K$ leads to higher robust accuracy, though at the cost of reduced time efficiency. In future work, we plan to improve the sampling speed of our defense by adopting the sampling acceleration methods.

**Selection of $t^*$.** An optimal choice of $t^*$ can improve robust accuracy, as the differences between benign and adversarial samples become indistinguishable beyond this point. As shown in Figure 1, selecting an appropriate $t^*$ provides a sweet spot that balances standard accuracy and robust accuracy. Nevertheless, even without fine-tuning or cherry-picking $t^*$, our baseline method still achieves competitive performance, demonstrating its robustness and stability.

## 5 CONCLUSION

In this paper, we present pioneering work on adversarial sampling and purification using denoising diffusion codebook models. On the attack side, our approach leverages adversarial noise selection from the codebook to achieve benign diffusion sampling, which not only enhances the quality of generated images but also improves attack success rates against state-of-the-art defenses. On the defense side, we propose perturbation-isolated codebook sampling, further strengthened by two complementary enhancements—blurring and sampling—which together enable more effective purification and consistently outperform existing diffusion-based purification methods. Extensive experiments on CIFAR-10 and ImageNet validate the effectiveness and scalability of our techniques. Overall, our study demonstrates that denoising diffusion codebook models deliver superior adversarial performance on both the attack and defense sides, providing a promising foundation for advancing adversarial robustness in diffusion models.

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

## A APPENDIX

### A.1 PROOF OF EQUATION 12

The proof of Equation 12 follows the DDCM original paper (Ohayon et al., 2025), where the selection of $\hat{k}$ should follow:

$$\hat{k}_t = \underset{1,...,K}{\arg\min} ||\mathcal{C}_t(k_t) - \sigma_t \nabla_{\boldsymbol{x}_t} \log p_f(y_a|\bar{\boldsymbol{x}}_{0,t})||^2 \tag{16}$$

$$= \underset{1,...,K}{\arg\min} ||\mathcal{C}_t(k_t)||^2 - 2\sigma_t \mathcal{C}_t(k_t) \cdot \nabla_{\boldsymbol{x}_t} \log p_f(y_a|\bar{\boldsymbol{x}}_{0,t}) + ||\nabla_{\boldsymbol{x}_t} \log p_f(y_a|\bar{\boldsymbol{x}}_{0,t})||^2 \tag{17}$$

$$= \underset{1,...,K}{\arg\min} ||\mathcal{C}_t(k_t)||^2 - 2\sigma_t \mathcal{C}_t(k_t) \cdot \nabla_{\boldsymbol{x}_t} \log p_f(y_a|\bar{\boldsymbol{x}}_{0,t}) \tag{18}$$

Ohayon et al. (2025) shows that $||\mathcal{C}_t(k_t)||^2$ is roughly equal for every $k$ and can be viewed as a constant; we have

$$\hat{k}_t = \underset{1,...,K}{\arg\min} ||\mathcal{C}_t(k_t)||^2 - 2\sigma_t \mathcal{C}_t(k_t) \cdot \nabla_{\boldsymbol{x}_t} \log p_f(y_a|\bar{\boldsymbol{x}}_{0,t}) \tag{19}$$

$$\approx \underset{1,...,K}{\arg\min} -2\sigma_t \mathcal{C}_t(k_t) \cdot \nabla_{\boldsymbol{x}_t} \log p_f(y_a|\bar{\boldsymbol{x}}_{0,t}) \tag{20}$$

$$= \underset{1,...,K}{\arg\max} \mathcal{C}_t(k_t) \cdot \nabla_{\boldsymbol{x}_t} \log p_f(y_a|\bar{\boldsymbol{x}}_{0,t}) \tag{21}$$

Therefore, the adversarial sampling for adversarial attack with DDCM can be formulated as Equation 12.

## A.2 THEORETICAL ANALYSIS ON BLURRING ENHANCEMENT

In this section, we give a theoretical analysis of the reason that blurring the input image leads to better adversarial purification results. We follow Li et al. (Li et al., 2024) for the theoretical analysis. Assume the probability of reversing the adversarial example $\hat{\boldsymbol{x}}_t$ to the clean example $\boldsymbol{x}_0$ using our blurring enhancement and DiffPure as $P(G)$ and $P(D)$, respectively. The $P(\cdot)$ can be calculated as $P(\cdot) = \int \mathbf{1}_{\{\boldsymbol{x}_0 \notin D_a\}} p(\boldsymbol{x}_0 | \hat{\boldsymbol{x}}_t) \mathrm{d}\boldsymbol{x}_0$, where the $D_a$ represents the set of adversarial examples. $p$ is the Gaussian distribution following Song et al. (Song et al., 2021b). As it's hard to give precise inference of $\hat{\boldsymbol{x}}_t$ after the blurring enhancement, we approximate the anti-aliased $\hat{\boldsymbol{x}}_t^{\mathrm{aa}}$ as $\hat{\boldsymbol{x}}_t^{\mathrm{aa}} \approx \psi \hat{\boldsymbol{x}}_t$, where $0 < \psi < 1$. The reason for this approximation lies in the blurring enhancement process, which can be seen as a direct reduction of pixel values by averaging pixels from the surrounding neighborhood using a mean filter (the filter weight is selected accordingly to ensure $0 < \psi < 1$). To further validate our findings, we recorded the values of $\hat{\boldsymbol{x}}_t - \hat{\boldsymbol{x}}_t^{\mathrm{aa}}$ during the experiments. Our analysis revealed that over $60\%$ of the pixels have positive values, while approximately $35\%$ of the pixels are nearly zero. Furthermore, more than $99\%$ of the images have a sum of absolute differences, $\sum |\hat{\boldsymbol{x}}_t - \hat{\boldsymbol{x}}_t^{\mathrm{aa}}| > 0$. Consequently, we can deduce that, for the majority of images, $\hat{\boldsymbol{x}}_t$ is greater than $\hat{\boldsymbol{x}}_t^{\mathrm{aa}}$.

We want to deduce:

$$P(G) > P(D),$$

wherein

$$\text{for } P(G) : p(\boldsymbol{x}_0 | \hat{\boldsymbol{x}}_t^{\mathrm{aa}}) \propto \exp\left(-\frac{\|\boldsymbol{x}_t^{\mathrm{aa}} - \sqrt{\bar{\alpha}_t}\boldsymbol{x}_0\|^2}{2(1 - \bar{\alpha}_t)}\right), \tag{22}$$

$$\text{for } P(D) : p(\boldsymbol{x}_0 | \hat{\boldsymbol{x}}_t) \propto \exp\left(-\frac{\|\boldsymbol{x}_t^{a} - \sqrt{\bar{\alpha}_t}\boldsymbol{x}_0\|^2}{2(1 - \bar{\alpha}_t)}\right). \tag{23}$$

**Proof:** According to the Bayes' rule,

$$
\begin{aligned}
p(\boldsymbol{x}_0 | \hat{\boldsymbol{x}}_t) &= p(\boldsymbol{x}_0) \frac{p(\hat{\boldsymbol{x}}_t | \boldsymbol{x}_0)}{p(\hat{\boldsymbol{x}}_t)} \\
&= p(\boldsymbol{x}_0) \frac{p(x_t | \boldsymbol{x}_0)}{p(\hat{\boldsymbol{x}}_t)} \\
&\propto p(\boldsymbol{x}_0) p(\boldsymbol{x}_t | \boldsymbol{x}_0)
\end{aligned}
\tag{24}
$$

As we assume we can recover $\boldsymbol{x}_0$ from the diffusion process with $\hat{\boldsymbol{x}}_t$ (for the infinite timestep, $\{\boldsymbol{x}_t\}_{t:0 \to T}$ and $\{\hat{\boldsymbol{x}}_t\}_{t:T \to 0}$ follow the same distribution (Song et al., 2021b)), we suggest $\hat{\boldsymbol{x}}_t^{\mathrm{aa}} := \boldsymbol{x}_t^{\mathrm{aa}}$ in our purification method and $\hat{\boldsymbol{x}}_t := \boldsymbol{x}_t^{a}$ in DiffPure. As we aim to prove blurring results in better purification, we assume that we use the same diffusion process for our method and DiffPure, i.e., $G$ and $D$ respectively. According to the reverse diffusion process,

$$
\begin{aligned}
P(G) &= \int \mathbf{1}_{\{\boldsymbol{x}_0 \notin D_a\}} p(\boldsymbol{x}_0 | \hat{\boldsymbol{x}}_t^{\mathrm{aa}}) \mathrm{d}\boldsymbol{x}_0 \\
&\propto \int \mathbf{1}_{\{\boldsymbol{x}_0 \notin D_a\}} p(\boldsymbol{x}_0) p(\boldsymbol{x}_t^{\mathrm{aa}} | \boldsymbol{x}_0) \mathrm{d}\boldsymbol{x}_0 \\
&\propto \int \mathbf{1}_{\{\boldsymbol{x}_0 \notin D_a\}} \exp\left(-\frac{\|\boldsymbol{x}_t^{\mathrm{aa}} - \sqrt{\bar{\alpha}_t}\boldsymbol{x}_0\|^2}{2(1 - \bar{\alpha}_t)}\right) p(\boldsymbol{x}_0) \mathrm{d}\boldsymbol{x}_0,
\end{aligned}
\tag{25}
$$

and

$$
\begin{aligned}
P(D) &= \int \mathbf{1}_{\{\boldsymbol{x}_0 \notin D_a\}} p(\boldsymbol{x}_0 | \hat{\boldsymbol{x}}_t) \mathrm{d}\boldsymbol{x}_0 \\
&\propto \int \mathbf{1}_{\{\boldsymbol{x}_0 \notin D_a\}} p(\boldsymbol{x}_0) p(\boldsymbol{x}_t^{a} | \boldsymbol{x}_0) \mathrm{d}\boldsymbol{x}_0 \\
&\propto \int \mathbf{1}_{\{\boldsymbol{x}_0 \notin D_a\}} \exp\left(-\frac{\|\boldsymbol{x}_t^{a} - \sqrt{\bar{\alpha}_t}\boldsymbol{x}_0\|^2}{2(1 - \bar{\alpha}_t)}\right) p(\boldsymbol{x}_0) \mathrm{d}\boldsymbol{x}_0,
\end{aligned}
\tag{26}
$$

We only need to compare $\|\boldsymbol{x}_t^{\mathrm{aa}} - \sqrt{\bar{\alpha}_t}\boldsymbol{x}_0\|^2$ and $\|\boldsymbol{x}_t^{a} - \sqrt{\bar{\alpha}_t}\boldsymbol{x}_0\|_2^2$ for comparing $P(G)$ and $P(D)$.

$$\|\boldsymbol{x}_t^{\mathrm{aa}} - \sqrt{\bar{\alpha}_t}\boldsymbol{x}_0\| \approx \|\boldsymbol{x}_t^a - \sqrt{\bar{\alpha}_t}\boldsymbol{x}_0 - (1 - \psi)\boldsymbol{x}_t^a\|, \tag{27}$$

Since $1 - \psi > 0$ always holds, thus we have:

$$\|\boldsymbol{x}_t^{\mathrm{aa}} - \sqrt{\bar{\alpha}_t}\boldsymbol{x}_0\| < \|\boldsymbol{x}_t^a - \sqrt{\bar{\alpha}_t}\boldsymbol{x}_0\|. \tag{28}$$

Hence,

$$\exp\left(-\frac{\|\boldsymbol{x}_t^{\mathrm{aa}} - \sqrt{\bar{\alpha}_t}\boldsymbol{x}_0\|^2}{2(1 - \bar{\alpha}_t)}\right)p(\boldsymbol{x}_0)\mathrm{d}\boldsymbol{x}_0 > \exp\left(-\frac{\|\boldsymbol{x}_t^a - \sqrt{\bar{\alpha}_t}\boldsymbol{x}_0\|^2}{2(1 - \bar{\alpha}_t)}\right)p(\boldsymbol{x}_0)\mathrm{d}\boldsymbol{x}_0, \tag{29}$$

which proves that using a blurred input image prior to diffusion-based purification can lead to a higher probability of purification to the corresponding clean image. However, it does not indicate that clipping the $\boldsymbol{x}_t$ smaller can result in better purification performance, as it can change the data distribution of the input image and lead to the wrong diffusion process.

### A.3 RELATED WORKS

#### A.3.1 ADVERSARIAL TRAINING

Adversarial training (AT) is one of the most practical approaches for improving a model's robustness against adversarial attacks. It trains the model with both benign and adversarial samples simultaneously. However, ensuring robustness against unseen attacks remains a major challenge, limiting the effectiveness of traditional adversarial training (Madry et al., 2018). To mitigate this, Gowal et al. (Gowal et al., 2021) and Rebuffi et al. (Rebuffi et al., 2021) introduced generated and augmented data to enhance generalization by increasing data diversity. Beyond data diversity, refining the objective formulation of AT has also shown promise. In particular, several methods have been proposed that explicitly consider model weights during adversarial training (Wu et al., 2020; Jin et al., 2023).

#### A.3.2 ADVERSARIAL PURIFICATION

Adversarial purification seeks to remove adversarial perturbations from adversarial examples without requiring the re-training of deep learning models, typically by exploiting the generative capabilities of advanced generative models. Early approaches based on generative adversarial networks (GANs) (Samangouei et al., 2018) and score-based matching models (Song et al., 2021b; Yoon et al., 2021) achieved state-of-the-art performance compared to adversarial training. With the rise of diffusion models, Nie et al. (Nie et al., 2022) demonstrated that diffusion-based adversarial purification methods surpass earlier techniques in restoring clean images. However, determining the optimal generation steps in diffusion-based purification remains a key challenge, and adversarial inputs can further disrupt the reverse generation process. To address these limitations, several recent works (Wang et al., 2022; Lee & Kim, 2023; Song et al., 2024; Bai et al., 2024) have introduced strategies to improve the robustness and effectiveness of adversarial purification.

### A.4 ADDITIONAL EXPERIMENTS

We provide additional experiments on CIFAR-10 in Tables 10 and 11. Furthermore, we evaluate the defense performance on the eyeglasses attribute of the CelebA-HQ dataset in Table 12, following the experimental settings of Nie et al. (2022).

### A.5 ABLATION STUDY ON ADVERSARIAL SAMPLING

**Selection of $K$.** The length of $\mathcal{C}$ is critical to the ASR of our adversarial sampling, as it determines the search space of adversarial examples (see Figure 2). Moreover, a larger codebook benefits image quality, since reconstructed adversarial examples are closer to the reference benign samples. However, increasing $\mathcal{K}$ significantly reduces time efficiency, leading to a trade-off between computational cost and attack performance.

**Selection of $t^*$.** The choice of $t^*$ controls the magnitude of adversarial sampling (see Figure 2). A larger $t^*$ allows for more DDCM-based adversarial sampling, which produces stronger adversarial

Table 10: **The defense performance against AutoAttack ($\ell_2 = 0.5$) on the CIFAR-10 dataset.**

| Method | Standard Accuracy(%) | Robust Accuracy(%) |
|---|---|---|
| WideResNet-28-10 | | |
| Rony et al. (2019) | 89.05 | 66.41 |
| Ding et al. (2020) | 88.02 | 67.77 |
| Rebuffi et al. (2021) | 87.50 | 78.32 |
| Nie et al. (2022) | 89.23 | 78.98 |
| Wang et al. (2022) | 84.85 | 75.28 |
| Song et al. (2024) | 92.10 | 81.52 |
| DDCM$_{\text{Pure}}$ | **93.85** | 80.82 |
| EDDCM$_{\text{Pure}}$ | 90.31 | **84.57** |
| WideResNet-70-16 | | |
| Rebuffi et al. (2021) | 88.54 | 80.86 |
| Gowal et al. (2021) | 88.74 | 74.03 |
| Nie et al. (2022) | 91.04 | 81.17 |
| Song et al. (2024) | 93.25 | 83.60 |
| DDCM$_{\text{Pure}}$ | **94.74** | 83.56 |
| EDDCM$_{\text{Pure}}$ | 92.65 | **86.63** |

Table 11: **The defense performance against C&W ($\ell_2 = 8/255$, EOT=50) on the CIFAR-10 dataset.**

| Method | Standard Accuracy(%) | Robust Accuracy(%) |
|---|---|---|
| WideResNet-28-10 | | |
| Nie et al. (2022) | 89.23 | 47.65 |
| Wang et al. (2022) | 84.85 | 21.71 |
| Song et al. (2024) | 92.10 | **89.91** |
| DDCM$_{\text{Pure}}$ | **93.85** | 85.70 |
| EDDCM$_{\text{Pure}}$ | 90.31 | 88.71 |

examples, but may also generate out-of-distribution samples with relatively lower image quality. Therefore, $t^*$ should be selected in accordance with the generative capacity of the diffusion model to balance attack strength and sample fidelity.

## A.6 VISUAL EXAMPLES

We give visual examples from our proposed adversarial sampling and adversarial purification, as shown in Figure 3. Although random sampling enhancement is applied in EDDCM$_{\text{Pure}}$, image consistency is preserved. However, the blurring enhancement may reduce the clarity of the purified image, which can be mitigated by incorporating additional deblurring models.

## A.7 LIMITATION

Despite the fact that adversarial sampling and purification with DDCM achieve promising performance compared with standard diffusion sampling in terms of adversarial robustness, the main limitation lies in time efficiency, as DDCM sampling currently does not support acceleration. In future work, we aim to enhance sampling efficiency by incorporating accelerated diffusion sampling into part of DDCM sampling. Another limitation is that adversarial performance is closely tied to the length of the codebook. However, enlarging the codebook requires substantial storage space. To address this, instead of storing the entire codebook at the initial state, we can store the codebook only at each diffusion timestep. While this approach prevents us from recording the codebook indices needed for image reconstruction, it is acceptable since our method is not designed for vector quantization tasks. By doing so, we can significantly improve space efficiency and further enhance the performance of the proposed sampling method.

Table 12: **The defense performance against BDPA+EOT ($\ell_{inf} = 16/255$) on the CelebA-HQ dataset.**

| Method | Defend Method | Standard Accuracy(%) | Robust Accuracy(%) |
|---|---|---|---|
| Richardson et al. (2021) | GAN | 93.95 | 75.00 |
| Nie et al. (2022) | Diffusion | 93.77 | 90.63 |
| Song et al. (2024) | Guided Diffusion | 94.52 | 93.81 |
| DDCM$_{Pure}$ | Codebook Diffusion | **95.08** | 93.54 |
| EDDCM$_{Pure}$ | Codebook Diffusion | 94.94 | **94.28** |

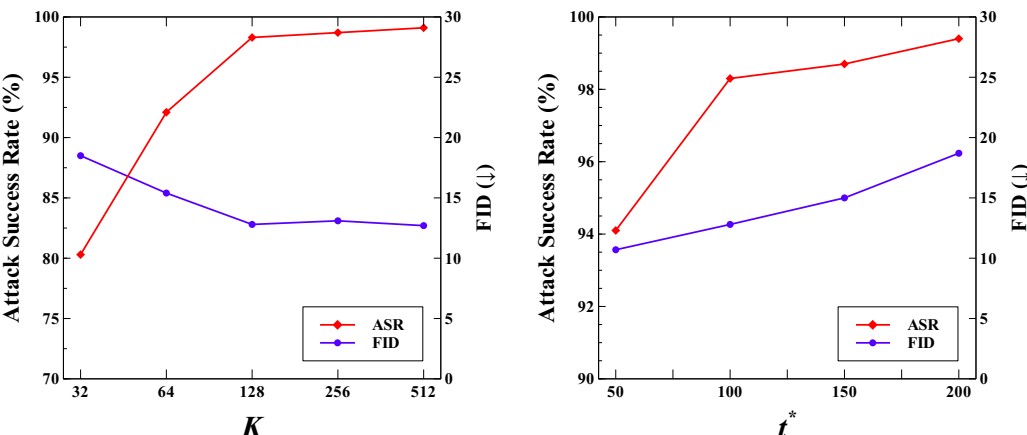

Figure 2: **The ablation study of parameters with DDCM$_{Adv}$ against ResNet50 on the ImageNet dataset.**

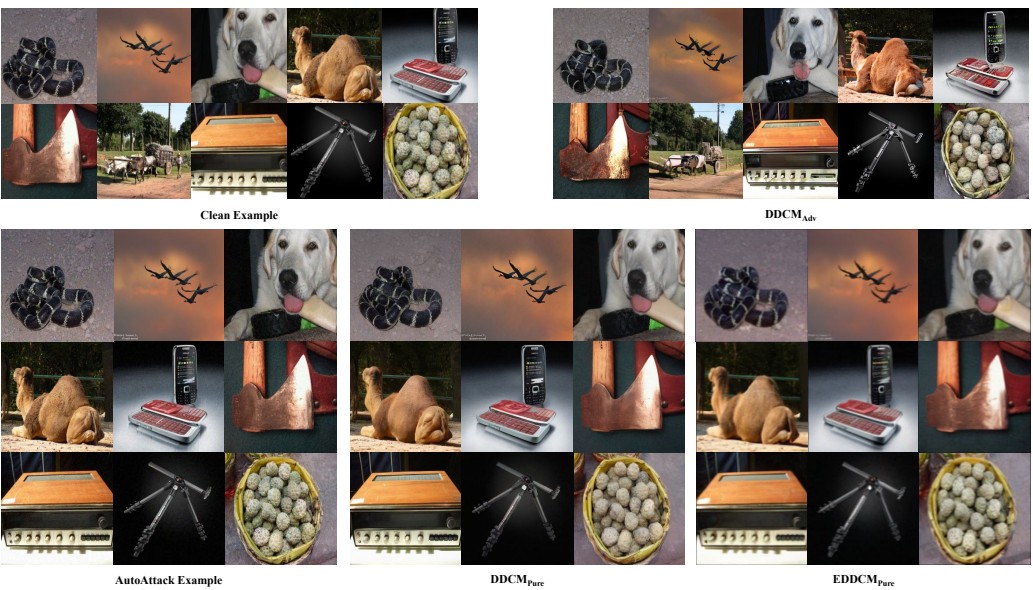

Figure 3: **Visual examples of proposed adversarial sampling and adversarial purification.**

