# OpenReview forum: "Adversarial Sampling and Purification with Denoising Diffusion Codebook Models"
_ICLR.cc/2026/Conference — ICLR 2026 Conference Withdrawn Submission_

### Official Review · Reviewer_E6UU · 2025-10-17

**Soundness:** 1
**Presentation:** 3
**Contribution:** 2
**Rating:** 4
**Confidence:** 5

**Summary:**

This paper proposes an adversarial sampling and purification method based on Denoising Diffusion Codebook Models. It highlights that traditional adversarial attacks and purification methods often degrade the quality and effectiveness of generative data in diffusion models. By introducing fixed codebook noise sampling, the proposed approach not only enhances the quality of adversarial samples but also increases attack success rates. Additionally, the purification framework isolates adversarial influences and strengthens purification through blurring and random noise sampling enhancements.

**Strengths:**

1. This paper is well-written and well-structured.
2. The paper is comprehensive, presenting both attack and defense methods.

**Weaknesses:**

1. The replacement of DDPM with DDCM lacks a reasonable motivation.
2. The method is unreasonable. In Line 4 of Algorithm 1, the authors select the noise with the highest similarity to the residual from the codebook. However, the optimal noise found in a discrete space is certainly not as effective as the noise found in a continuous space. Additionally, there is an issue with the formulation of Equation 5; the vector dot product has a specific formula that should be used.
3. The experimental comparison is unreasonable. To validate the effectiveness of the method, it should align with previous methods [1,2,3] and use adaptive attacks for testing. I also don't believe the method is effective because previous papers have concluded that diffusion-based diffusion works effectively due to the randomness of noise added internally. The discrete noise codebook clearly lacks the randomness found in continuous space.
[1] Robust evaluation of diffusion-based adversarial purification
[2] Diffusion models demand contrastive guidance for adversarial purification to advance
[3] Diffusion models for adversairal purification
[4] Towards Understanding the Robustness of Diffusion-Based Purification: A Stochastic Perspective

**Questions:**

No

---

### Official Review · Reviewer_Dvgv · 2025-10-20

**Soundness:** 1
**Presentation:** 3
**Contribution:** 2
**Rating:** 4
**Confidence:** 2

**Summary:**

This paper proposes using Denoising Diffusion Codebook Models (DDCMs) for both adversarial attack ($DDCM_{Adv}$) and adversarial purification defense ($DDCM_{Pure}$ / $EDDCM_{Pure}$). The core idea is to replace the random noise sampling in standard diffusion models with noise selection from a predefined codebook. The authors claim this approach "preserves the benign data distribution" during attack and achieves "adversarial-isolated sampling" for defense. The paper reports state-of-the-art (SOTA) robust accuracy on CIFAR-10 and ImageNet.

**Strengths:**

- Reported SOTA Performance: The proposed defense ($EDDCM_{Pure}$) reports SOTA robust accuracy on standard benchmarks (CIFAR-10, ImageNet) against various attacks, including AutoAttack, outperforming strong baselines like DiffPure and MimicDiffusion.

- Time Efficiency: Compared to Guided Backward (GB) methods that require gradient computation via backpropagation (e.g., MimicDiffusion), the proposed defense demonstrates a significant time efficiency advantage on large-scale datasets like ImageNet.

- Pioneering Application: To our knowledge, this is the first work to apply the Denoising Diffusion Codebook Model (DDCM) paradigm specifically to the task of adversarial purification, exploring a new model variant for this domain.

- Attack Image Quality: The $DDCM_{Adv}$ attack method generates adversarial examples that achieve excellent scores on image quality metrics such as FID and BRISQUE.

**Weaknesses:**

- Unfair Attack Evaluation Setup

The attack evaluation in Table 1 mixes two distinct attack methods: unconstrained generative attacks (AdvDiff, $DDCM_{Adv}$) and $L_p$-bounded perturbation attacks (AutoAttack). More confusingly, the authors state that "the clean examples for AutoAttack and PGD are images generated by LDM". This comparison may unfair and non-standard. The evaluation should be conducted under a unified, standard task setting (e.g., starting from the same original benign images) to allow for a fair assessment of $DDCM_{Adv}$'s relative performance.

- (Critical) Questionable Validity of SOTA Results: Gradient Masking and Flawed Baselines

SOTA May Stem from Gradient Masking: The SOTA results of $EDDCM_{Pure}$ may be a result of gradient masking, not true robustness. The defense relies heavily on non-differentiable or gradient-stopping operations, namely Blurring and argmax (for codebook selection). The authors themselves note these operations have "zero gradients", a classic indicator of gradient masking. One piece of supporting evidence is that the PGD+EOT results (Tables 5 & 6) are suspiciously close to the AutoAttack results (Tables 3 & 4) for $DDCM_{Pure}$ and $EDDCM_{Pure}$. This may suggest that the defense is breaking gradients, causing gradient-based adaptive attacks to fail.

Untrustworthy AutoAttack Baseline: The paper uses DiffPure as a key SOTA baseline. However, recent work has pointed out that the reported AutoAttack results for DiffPure are severely overestimated. This stems from two distinct bugs: (1) a bug in DiffPure's own adjoint method[1] that causes incorrect gradient computation (a non-AutoAttack-related gradient masking issue), and (2) a known bug in AutoAttack itself when evaluating defenses that involve randomness and final sample selection (an AutoAttack-related evaluation bug) [2]. Since this paper replicates the same baseline setup as DiffPure, the reported SOTA comparisons against it are untrustworthy.

- (Critical) Internal Contradictions in Methodology

Attack Contradiction: The paper claims $DDCM_{Adv}$ "preserves the benign data distribution". However, its core mechanism (Eq. 12) and derivation (Appendix A.1) clearly show it is an  gradient-guided attack, merely a discrete approximation of standard gradient guidance.

Defense Contradiction & Randomness: The paper claims its defense achieves "adversarial-isolated sampling". However, the baseline $DDCM_{Pure}$ (Eq. 14) explicitly uses the adversarial example $\hat{x}$ as guidance. More fundamentally, if the conclusion from ADDT [2] (that the robustness of diffusion purification comes primarily from its randomness) is correct, then the DDCM approach, which replaces continuous Gaussian random sampling with discrete codebook selection, actually limits this randomness. This should, in theory, reduce robustness. The paper's SOTA results are in direct conflict with this theoretical implication, a contradiction that is never discussed. The SOTA $EDDCM_{Pure}$ method (Alg. 2) abandons this adversarial guidance after $t^*$ and reverts to random sampling. This design choice may suggest that the adversarial guidance from Eq. 14 is more detrimental in the latter half of the process, which seems to undermine the paper's core  of "adversarial isolation."

- Flawed Theoretical Claim

The proof for Theorem 1 ("Blurring Enhancement") appears non-rigorous. It incorrectly assumes that $\psi < 1$ implies $|| \psi \hat{x}_t - c || < || \hat{x}_t - c ||$ in a vector space, which is not generally true.

- Limited Novelty

The contribution of this paper appears "paradigmatic." Applying a new diffusion variant (like DDCM) to the purification task is rapidly becoming a common pattern[3][4].

The key components of $EDDCM_{Pure}$ lack novelty: (1) Blurring is an ancient (and generally considered ineffective) defense technique. (2) The selection of $t^*$ (including randomization) has already been discussed in detail in the original DiffPure paper. The main contribution seems to be combining these known components with DDCM, which is of limited conceptual novelty.

- Lack of Reproducibility

Given the high prevalence of gradient masking and evaluation pitfalls (like the AA bug) in diffusion-based purification, the authors have not provided source code. This makes it impossible for reviewers to independently verify the results' correctness or to design stronger adaptive attacks to test the true robustness of the method. This severely diminishes the paper's credibility.

[1]Towards Understanding the Robustness of Diffusion-Based Purification: A Stochastic Perspective.

[2]Robust evaluation of diffusion-based adversarial purification.

[3]Consistency Purification: Effective and Efficient Diffusion Purification towards Certified Robustness.

[4]Wang J, Lyu Z, Lin D, et al. Guided diffusion model for adversarial purification[J]. arXiv preprint arXiv:2205.14969, 2022.

**Questions:**

- Adaptive Self-Attack: What is the performance of the $EDDCM_{Pure}$ defense when attacked by the paper's own $DDCM_{Adv}$? This is a basic and necessary adaptive attack test that is missing from the evaluation.

- $DDCM_{Adv}$ Attack Metrics:Why does the paper only report image quality (FID, etc.) for $DDCM_{Adv}$  and not the $L_\infty$ or $L_2$ distance between the generated adversarial examples and the original benign images?

- Why is there such a large performance gap in ASR for $DDCM_{Adv}$ when attacking ResNet50 (98.3%) vs. ViT-B (84.3%)?

- Blur Kernel Choice: The paper uses a very unusual blur kernel: [[1, 1, 1, 1, 1], [1, 1, 0, 1, 1], [1, 1, 1, 1, 1]]. Its shape (non-symmetric) and the zero in the center are highly non-standard. What is the justification for this specific kernel? Was it tuned to maximize defense performance (or, potentially, gradient masking effects)?

- Ablation Study on $EDDCM_{Pure}$: Why does $EDDCM_{Pure}$ (Alg. 2) switch to random codebook selection after $t^*$? A key ablation is missing: what is the performance if adversarial guidance (Eq. 14) is used for all steps $T \rightarrow 1$ without blurring enhancement?

- Codebook Memory Overhead: What is the practical RAM overhead of this method? The model appears to require storing a codebook $\mathcal{C}_t = [z_t^{(1)}, ..., z_t^{(K)}]$ for every timestep $t$. For a large $K$ (e.g., 512, from Figure 1) and many diffusion steps, does this not result in an impractically large memory footprint?

- Typo in Equation 9? Equation 9, $x_{t-1}=\mu_{\theta}(x_{t},t)+\sigma_{\theta}z_{t}\nabla_{x}+log~p(\hat{x}|x_{t};t)$, appears to have a typo. The placement of the + sign after $\nabla_x$ is strange. Is this intended to be a product, or should the $\nabla_x$ apply to the $log$ term as in Equation 10?

---

### Official Review · Reviewer_9nWy · 2025-10-28

**Soundness:** 3
**Presentation:** 3
**Contribution:** 3
**Rating:** 4
**Confidence:** 4

**Summary:**

This paper investigates the use of Denoising Diffusion Codebook Models (DDCMs) for both creating and defending against adversarial examples. The authors propose an adversarial sampling technique that, instead of adding perturbations, generates attacks by altering the codebook selection process during the reverse diffusion, guided by a target classifier. For adversarial purification, the paper introduces a framework that also leverages this codebook mechanism, using the adversarial sample as a reference to guide the generation. The authors also describe two enhancements to this purification: a preprocessing step that blurs the input  and a modification to the sampling process that switches to random codebook selection after a specific timestep $t^*$.

**Strengths:**

I find the paper's focus on Denoising Diffusion Codebook Models (DDCMs) for adversarial robustness to be a new direction. Exploring how this specific architecture, which replaces random noise sampling with selections from a predefined codebook, behaves in both attack and defense scenarios is a timely investigation.

The proposed adversarial attack is an interesting approach to generating unrestricted examples. Instead of directly adding gradient-based perturbations to the sampling process , the method works by modifying the *selection* of the codebook entries to steer the generation toward a target class. This seems to help in generating attacks that maintain high image quality , as suggested by the metrics in Table 1.

I think the paper also does a thorough job in developing its purification framework. The authors present a baseline method  and an enhanced version (EDDCM) that incorporates a blurring preprocessing step and a random sampling phase after a certain timestep $t^*$. The experimental results presented in Tables 3, 4, 5, and 6 show that this defense framework achieves high robust accuracy against several strong attacks, including adaptive ones.

**Weaknesses:**

My main concern with this paper lies in the evaluation of the proposed defense. The reported robust accuracy figures, such as 86.40% against AutoAttack and 86.28% against BPDA on CIFAR-10, are exceptionally high and, in my opinion, highly suspicious. First, **recent work suggests that the true robustness limit on CIFAR-10 (even for humans) is likely much lower than 86% due to artifacts in the attack generation process itself [5]**. Second, **the field of diffusion-based purification has a history of defenses reporting very high numbers that are subsequently broken by stronger, specifically-designed adaptive attacks [1, 2, 3, 6]**.

The proposed enhanced defense, $EDDCM_{Pure}$, relies on two components that are classic sources of gradient obfuscation: a non-differentiable blurring filter (BF) and a stochastic random codebook selection for $t < t^*$ [4]. While the authors do test against BPDA and PGD+EOT, it is not clear if these attacks are powerful enough to bypass this specific combination of obfuscation techniques. A defense this strong, especially one based on non-differentiable and stochastic components, requires a much more convincing and rigorous adaptive evaluation to rule out a false sense of security [1, 4]. I would expect to see a new adaptive attack designed to specifically overcome both the blurring and the randomized sampling, as has been done for other diffusion defenses [2, 3].

Another point is the practical limitation of the method, which the authors do touch upon. The performance of both the attack and defense is heavily tied to the codebook size $K$, and the DDCM sampling process itself is noted to be slow and currently incompatible with acceleration methods. The ablation in Figure 1 clearly shows that high robustness requires a large $K$, which in turn incurs a significant time cost. This trade-off between robustness and efficiency could be a major barrier to practical adoption.

Finally, the paper feels a bit like two separate contributions—an attack and a defense—that happen to use the same DDCM primitive. The connection between them is not fully exploited. For example, the proposed $DDCM_{Adv}$ attack is evaluated against existing defenses in Table 2, while the proposed $EDDCM_{Pure}$ defense is evaluated against existing attacks in Tables 3-6. A crucial experiment seems to be missing: evaluating the proposed defense directly against the proposed attack. This would provide a clearer picture of their relative strengths and weaknesses.

### References
[1] Lee, Minjong, and Dongwoo Kim. "Robust evaluation of diffusion-based adversarial purification." arXiv preprint arXiv:2303.09051 (2023).
[2] Kang, Mintong, Dawn Song, and Bo Li. "DiffAttack: Evasion Attacks Against Diffusion-Based Adversarial Purification." Advances in Neural Information Processing Systems 36 (2023).
[3] Chen, Huanran, et al. "Robust Classification via a Single Diffusion Model." In International Conference on Machine Learning (ICML), 2024.
[4] Athalye, A., Carlini, N., Wagner, D. "Obfuscated gradients give a false sense of security: Circumventing defenses to adversarial examples." In International Conference on Machine Learning (ICML), PMLR, 2018.
[5] Bartoldson, Brian R., James Diffenderfer, Konstantinos Parasyris, and Bhavya Kailkhura. "Adversarial Robustness Limits via Scaling-Law and Human-Alignment Studies." In International Conference on Machine Learning (ICML), 2024.
[6] Liu, Yiming, et al. "Towards Understanding the Robustness of Diffusion-Based Purification: A Stochastic Perspective". ICLR, 2025.

**Questions:**

My main questions for the authors revolve around the evaluation of the defense, which I find insufficient to support the exceptionally strong claims of robustness. The reported 86.4% robust accuracy on CIFAR-10 is a major outlier that requires extraordinary evidence, especially given that recent studies suggest the practical robustness limit on this dataset may be much lower [5].

1.  My primary concern is the potential for **obfuscated gradients**, a common issue for defenses that report such high numbers [4]. The enhanced defense ($EDDCM_{Pure}$) combines two mechanisms known to cause this: (1) a non-differentiable blurring filter (BF) as preprocessing  and (2) stochastic random codebook selection for $t < t^*$. While BPDA and PGD+EOT are tested, these are often insufficient for bypassing such defenses. Could the authors please design and evaluate a stronger adaptive attack that *specifically* targets both of these components? For instance, an attack that attempts to approximate the gradient of the blurring filter and also optimizes over the expectation of the random codebook selection, perhaps by attacking the deterministic, reference-guided part of the sampling ($t > t^*$)  more aggressively.

2.  Related to the evaluation, the baselines for purification methods seem to be based on older evaluations. For example, DiffPure's  robustness is known to be much lower than originally reported when evaluated under strong, specific adaptive attacks [1, 2, 3]. Could the authors please re-evaluate the key baselines (like DiffPure ) using more recent and effective adaptive attacks to provide a more realistic comparison point for your own method?

3.  I noticed a seemingly critical experiment is missing. The paper introduces both a novel attack ($DDCM_{Adv}$)  and a novel defense ($EDDCM_{Pure}$). However, the defense is only tested against *existing* attacks (like AutoAttack), and the attack is only tested against *existing* defenses. How does the proposed $EDDCM_{Pure}$ defense perform when evaluated directly against the proposed $DDCM_{Adv}$ attack? This seems essential for understanding the robustness properties within the DDCM framework.

4.  Could you clarify the practical cost of the best-performing defense? The ablation in Figure 1  shows robustness scaling with codebook size $K$, which also increases time. What value of $K$ was used to achieve the 65.03% robustness on ImageNet (Table 4 ) and the 86.40% on CIFAR-10 (Table 3 )? What is the precise wall-clock time for a single image purification with this $K$? This is important for understanding the practical trade-off.

---

### Official Review · Reviewer_jYdH · 2025-10-31

**Soundness:** 3
**Presentation:** 2
**Contribution:** 2
**Rating:** 2
**Confidence:** 4

**Summary:**

This paper proposes adversarial attack and defense methods using Denoising Diffusion Codebook Models (DDCMs). The key insight is leveraging DDCM's fixed codebook sampling to maintain benign data distribution instead of adversarial distribution. For attacks, adversarial examples are generated via conditional codebook selection. For defense, the method isolates adversarial influence through codebook-based purification enhanced with blurring and random sampling. Experiments on CIFAR-10 and ImageNet demonstrate state-of-the-art performance in both attack and defense scenarios, though with notable computational costs.

**Strengths:**

1. This appears to be the first work systematically investigating adversarial attacks and defenses using Denoising Diffusion Codebook Models (DDCMs), opening a new research direction in adversarial robustness.
2. The paper addresses both attack and defense, providing a complete picture of DDCM's role in adversarial robustness.

**Weaknesses:**

1. The paper is difficult to read and have no figures to help readers intuitively understand the motivation and methodology. Visual illustrations would greatly improve comprehension.

2. In Algorithm 2, line 9 have `obtains inference of predicted` $\bar{x}_{0,t}$ , but this is not used in subsequent steps. Why is this operation still necessary? In line 10, noise is randomly selected from the codebook without depending on the model output at all. Why doesn't this lead to generation collapse? This is unreasonable, and the authors need to provide corresponding explanations and experimental analysis.

3. In Line 235, the claim that $\frac{\partial}{\partial t} D_{KL}(p_t||q_t) = 0$ at $x_T$ is stated but not proven in the context of codebook sampling.

4. The authors note that `in-place operations with zero gradients provide defense`, but this suggests potential gradient obfuscation rather than true robustness. Codebook selection can also be attacked by approximately differentiable operations, which the authors have not considered, making the comparison unfair. More sophisticated adaptive attacks that specifically target the codebook selection mechanism should be considered.

**Questions:**

The mean filter specification [[1,1,1,1,1],[1,1,0,1,1],[1,1,1,1,1]] appears unusual with a zero center - this should be explained or corrected.

---

### Note · Authors · 2025-11-12

I have read and agree with the venue's withdrawal policy on behalf of myself and my co-authors.